# Comparison of Hormonal, Inflammatory, Muscle Damage and Oxidative Stress Biomarkers Changes in Response to High-Intensity Interval, Circuit and Concurrent Exercise Bouts

**DOI:** 10.3390/sports13060184

**Published:** 2025-06-12

**Authors:** Francisco Javier Martínez-Noguera, Linda H. Chung, Amelia Guadalupe-Grau, Silvia Montoro-García, Pedro E. Alcaraz

**Affiliations:** 1Research Center for High Performance Sport, Catholic University of Murcia (UCAM), Guadalupe, 30107 Murcia, Spain; palcaraz@ucam.edu; 2GENUD Toledo Research Group, Faculty of Sport Sciences, University of Castilla-La Mancha, 45071 Toledo, Spain; amelia.guadalupe@uclm.es; 3CIBER on Frailty and Healthy Aging (CIBERFES), Instituto de Salud Carlos III, 28029 Madrid, Spain; 4Grupo Mixto de Fragilidad y Envejecimiento Exitoso UCLM-SESCAM (TEC2022-007), Instituto de Investigación Sanitaria de Castilla-La Mancha (IDISCAM), Junta de Comunidades de Castilla-La Mancha (JCCM), 45071 Toledo, Spain; 5Preclinical Research of Bioactive Compounds and Drugs (PREBIOF), Izpisúa Lab HiTech, Faculty of Health Sciences, Catholic University of Murcia (UCAM), Campus los Jerónimos, 30107 Murcia, Spain; smontoro@ucam.edu

**Keywords:** resistance training, testosterone, myostatin, IL-6, TNF-α

## Abstract

Purpose: Although chronic resting hormonal changes were traditionally considered to modulate muscle tissue remodeling and growth, our knowledge of exercise on the acute post-exercise hormonal response is limited. Moreover, the type of exercise protocol may trigger different hormonal profiles. The aim of this study was to evaluate changes in muscle damage, as well as hormonal and inflammatory markers following the response to three different resistance training protocols. Methods: A crossover study was conducted in which 33 recreationally active men were randomly assigned to three different training groups: high-intensity interval training (HIIT), concurrent training (CT), and high-intensity resistance circuit (HRC) training. Blood biomarkers were measured by standard procedures at rest, after exercise (P0), 30 min (P1), 24 h (P24), and 48 h (P48) after exercise. Results: Regarding testosterone, the Friedman test detected a significant time × group interaction (*p* = 0.004), and Durbin–Conover showed higher levels in HRC compared to HIIT at P1 (*p* = 0.006) and P48 (*p* = 0.021). However, CT showed higher levels than HIIT (*p* = 0.008) at P1. Concerning myostatin, there was a trend in the time × group interaction (*p* = 0.056) with lower values in HRC compared to CT in P1 (*p* = 0.003), and a trend between HRC and HIIT in P1 (*p* = 0.056). Conclusions: HRC generates higher levels of testosterone than HIIT in the acute (P1) and late (P48) phases of recovery and produces lower levels of myostatin than CT and HIIT (P1) in the acute phase of recovery.

## 1. Introduction

The endocrine system’s expression is a major contributor to evoking exercise-related physiological and biochemical adaptations [1]. These adaptations are linked to the secretion of various hormones, such as testosterone (T) and cortisol (C) [1]. There is strong evidence that T plays a pivotal anabolic (androgenic) role in various tissues, such as muscle, bone, and red blood cells, but it is also involved in growth [2]. However, C is also fundamental to catabolic pathways, mobilizing carbohydrates, fats, and protein for energy production, as well as suppressing the immune system [3].

Resistance exercise can modify hormone concentrations, like T, insulin-like growth factor-1 (IGF-1), and growth hormone (GH), due to mechanical overload, resulting in increased muscle mass [4]. The acute elevations in circulating hormone concentrations are due to a combination of increased secretion, reduced hepatic clearance, lower plasma volume or decreased degradation rates, and interaction with cellular membrane/cytoplasmic/nuclear receptors of the skeletal muscle [5]. Kraemer and Ratamess (2005) recognize the critical role played by the pattern of hormonal factors coupled with mechanical stimulation of intramuscular signaling, promoting a sequence of molecular processes that lead to the adaptive response of the muscle, mainly hypertrophy [5].

When there is an increase in oxygen consumption mediated by physical exercise, there is an increase in the production of oxygen-derived free radicals (i.e., reactive oxygen species; ROS), with an important signaling function [6]. Recently, there has been renewed interest in the hormesis behavior of ROS. The classic U-shaped curve of hormesis, initially described in relation to toxins, has also been used to analyze how ROS, aging, and physical exercise interact [7]. It demonstrates that both under-activity and over-exercise (overtraining) can be detrimental, but that moderate exercise has significant benefits: it reduces oxidative stress, decreases the risk of various diseases, and contributes to a longer and healthier life [7]. Extensive research has shown that, at low levels, ROS produce a beneficial effect on health, but at high levels and over a prolonged period, they can cause cellular damage [8,9]. The redox response has already been tested following exercise by evaluating the changes in thiobarbituric acid-reactive substances (TBARS) [10]. Furthermore, Scheele et al. [11] showed the relationship between ROS and exercise responses, hypothesizing that the production of superoxide and hydrogen peroxide free radicals from muscle mitochondria is key to the induction of proinflammatory cytokines, such as myokines.

In parallel, inflammatory markers are also upregulated following exercise-induced muscle damage, which is essential for muscle repair and regeneration to occur [12]. Intramuscular inflammation is a dynamic and effective coordinated process that ultimately promotes adaptive remodeling and a return to homeostasis [13]. Two of the most studied inflammatory cytokines during and after exercise are interleukin-6 (IL-6) and tumor necrosis factor α (TNF-α) [14]. Plasma levels of IL-6 can increase up to 100-fold after acute exercise (i.e., cycling), and circulating levels of muscle-derived IL-6 are closely related to exercise duration and intensity [15,16]. In addition, IL-6 is known to stimulate TNF-α expression [17].

Other biomarkers that are also linked to stress during exercise and recovery are creatine kinase (CK) and myostatin (MYO). CK is a marker of muscle damage and increases in parallel with IL-6, depending on the volume and intensity of exercise [18]. However, MYO is known to control muscle growth and repair by inhibiting the proliferation and differentiation of the satellite cells (i.e., muscle stem cells) [19]. Therefore, studies over the past two decades have provided important information on this cascade of physiological events that prepares the body for the subsequent exercise session and mediates the recovery and long-term adaptations that will lead to better performance. In a study evaluating the effects of three HIIT protocols (15 s/15 s, 30 s/30 s, and 60 s/60 s) on markers of muscle damage, it was found that all protocols significantly increased CK, myoglobin, and lactate dehydrogenase (LDH) after exercise. However, the greatest differences between protocols were observed in myoglobin levels, especially in 15 s/15 s and 60 s/60 s protocols [20]. However, in another study comparing aerobic training at 60% of VO_2MAX_, aerobic training at 80% of VO_2MAX_, a resistance exercise session with a bi-set protocol, and a resistance exercise session with a multiple sets protocol, it was seen that aerobic exercise at 80% of VO_2MAX_ generated a significant increase in CK at 24 h compared to the bi-set resistance exercise protocol. However, the endurance sessions (bi-set and multiple sets) elicited greater immediate increases in LDH than the aerobic protocols [21].

When it comes to improving athletic performance, coaches can choose from a variety of exercise training methods. Among them, concurrent training (CT) (i.e., combination of strength and endurance training) has been shown to improve maximal oxygen uptake (VO_2MAX_) but inhibit strength development compared to strength training alone [22]. On the other hand, high-intensity interval training (HIIT) uses repeated, short-to-long duration, high-intensity exercise intervals at or above the maximum steady-state lactate rate, interspersed with active (e.g., light exercise) or inactive recovery periods [23]. HIIT has been demonstrated to improve VO_2MAX_ and anaerobic power and reduce body weight, fat mass, and fasting glucose compared to control or other types of training [24,25]. In addition, high-intensity resistance circuit training (HRC) is a method that uses high loads with short rests between sets [26] and has been observed to increase maximal strength and aerobic capacity compared to traditional strength training [27,28].

However, very little is currently known about the difference in the modulation of the endocrine system in the different types of training methodologies. This indicates a need to understand the degree of hormonal changes that exist among the types of training protocol, bearing in mind that a negative hormonal profile (↓T, ↓GH, ↑C) can unfavorably affect muscle recovery [29].

Therefore, the objective of this study was to evaluate the responses of hormonal (T, GH, C, and IGF-1), inflammatory (IL-6 and TNF-α), muscle damage-atrophy (CK, MYO) biomarkers in both the acute and recovery phase after CT, HRC, and HIIT in recreationally active young men. We hypothesized that the HRC training protocol would generate a better anabolic environment due to a greater increase in T, GH, and IGF-1 compared to the HIIT and HRC training protocols.

## 2. Material and Methods

### 2.1. Study Design

A randomized, crossover, counterbalanced study was employed. The software Randomizer [30] was used to divide the participants randomly into 3 groups: high-intensity interval training (HIIT), concurrent training (CT), and high-intensity resistance circuit training (HRC). Between the different types of training, there were 5 washout days with rest (i.e., no resistance training was performed).

### 2.2. Participants

Thirty-three healthy, non-smoking, recreationally active males (150 to 300 min of moderate-intensity activity or 75 to 150 min of vigorous-intensity activity per week, plus muscle-strengthening activities 2 or more days per week) participated in this study. Subjects were recruited from the University. The inclusion criteria were as follows: (a) regularly practice physical activity (3 times a week), (b) have a body mass index (BMI) > 20 and <27 kg·m^−2^, (c) able to complete at least 20 min of intense workout, and (d) have a VO_2MAX_ > 35 mL·kg^−1^·min^−1^. Volunteers were excluded if they were under 18 years of age, had cardiovascular, respiratory, metabolic, or other significant neurological or orthopedic pathologies limiting exercise, and/or had any other medical condition contraindicating exercise. First, participants were informed about the procedures and signed informed consent. The study was conducted according to the guidelines of the Helsinki Declaration for Human Research [31] and was approved by the Ethics Committee of the Universidad Católica de Murcia (CE031704).

### 2.3. Procedures

Participants came to the UCAM Research Center for High Performance Sport five times. At visit 1, a medical examination and blood analysis were performed to check for health status and to verify compliance with the inclusion criteria. In visit 2, a VO_2MAX_ test was performed to determine whether the subjects were able to withstand an intense workout of at least 20 min of duration. This was followed by a familiarization session where the subjects performed the tests to be used for evaluation, as well as the exercises of the training protocols, to prevent learning effects during the experimental phase. At visits 3, 4, and 5, one of the three training protocols (HIIT, CT, and HRC) was carried out, where venous blood collection was performed at baseline, at 0 min (P0), after 30 min (P1), after 24 h (P24) and after 48 h (P48) post-exercise. All groups performed the training in a similar time frame (9:00–11:00 a.m.) to avoid interference of circadian rhythms in the results.

### 2.4. Medical Examination

A medical examination, performed by the research center’s physician, included a health history, resting electrocardiogram, and examination (auscultation, blood pressure, etc.) to confirm that the volunteer was healthy enough to participate in the study.

### 2.5. Blood Samples

Blood samples were collected after an overnight fast (12 h) from each participant between 8:30 a.m. and 10:30 a.m. A total of 6.5 mL of peripheral blood was collected aseptically from an antecubital vein for blood analysis, one in a 3 mL tube with ethylenediaminetetraacetic acid (EDTA) and one in a 3.5 mL tube with polyethylene terephthalate (PET). Erythrocyte counts were performed from whole blood on a Cell-Dyn 3700 automated analyzer (Abbott Diagnostics, Chicago, IL, USA) using internal (Cell-Dyn 22) and external (Program of Excellence for Medical Laboratories-PEML) controls. Erythrocytes, hemoglobin, haematocrit, and haemacytometry indices were determined. These data served to verify the health status of the subjects and were not included in the study data.

In addition, venous blood samples were taken at baseline, P0, P1, P24, and P48 to measure T, C, IGF1, GH, IL-6, TNF-α, CK, MYO, and thiobarbituric acid reactive substances (TBARS) (visits 3, 4 and 5). This was performed with the following criteria: after eight hours of sleep, 12 h of fasting, and three days of rest (i.e., no exercise). Venous blood was collected in Vacutainer^®^ tubes with EDTA and serum and centrifuged at 3500 rpm for 15 min at room temperature to obtain platelet-poor plasma. Aliquots of plasma and serum obtained were stored at −80 °C to allow analysis of hormones, lipid profile, and ELISA batches.

### 2.6. Incremental Test

A maximal incremental test was performed on a treadmill (Technogym Run Excite Med., Cesena, Italy) with a metabolic cart (Metalyzer 3B. Leipzig, Germany) to determine peak running VO_2_. The test started with a 5 min warm-up at 5.0 km·h^−1^. Afterwards, the speed was brought up to 9 km·h^−1^ for 2 min which was increased by 1 km·h^−1^ increments every 2 min until physiological exhaustion or until a RER > 1.05 was reached [32]. To confirm VO_2MAX_, at least 2 of the following conditions had to be achieved: established plateau in the final VO_2_ values (increase ≤2.0 mL·kg^−1^·min^−1^ in the last 2 loads), reached maximum theoretical heart rate (HR) ((220 − age) − 0.95), RER ≥ 1.15 or lactate ≥ 8.0 mmol·L^−1^ [33].

### 2.7. Acute Exercise Training Protocols

Before all the training protocols a standardized and specific warm-up was performed. The work density was controlled to be 1:3 across groups (HIIT (total 32 min) 5 s effort, 15 s recovery; HRC (total 45 min) 15 s effort, 45 s recovery; CT (total 110 min).

HIIT protocol: Participants performed 2 blocks of 3 sets of 6 repetitions of 20 m sprints in an indoor court. Fifteen seconds of rest was given between repetitions, 3 min of rest between sets and 5 min of rest between blocks. Verbal encouragement was given throughout the all-out sprints.

HRC protocol: Participants performed 2 circuits of 3 sets of 3 exercises [28]. The first circuit consisted of lat pull-down, knee extension, and chest press. The second circuit contained knee flexion, elbow flexion, and sitting calf raise. All exercises were performed at a pre-determined 6-RM with 40 s of rest between exercises. Five min of rest was provided between circuits. The participants were instructed to perform the concentric phase at maximum velocity and to control the eccentric phase to last 3 s. Verbal encouragement was given throughout the exercises, as well as monitoring of correct execution.

CT protocol: The traditional CT consisted of a strength block, followed by an aerobic block. Participants performed 3 sets of 6 exercises in the strength block, which were lat pull-down, knee extension, chest press, knee flexion, elbow flexion, and sitting calf raise. All exercises were performed at a pre-determined 6-RM, and 3 min of rest was given between sets. The concentric–eccentric ratio was performed the same as HRC. After 5 min of rest following the strength block, participants ran on a treadmill for 20 min with a 1% grade and at a speed that marked their second ventilatory threshold (previously determined). The day prior to the training visits, subjects followed a standardized diet that included 8.8 g/kg body weight (BW) of carbohydrates, 1.4 g/BM of protein, and 0.8 g/BM of fat. In addition, they were instructed to consume breakfast 2 h before each training visit, consisting of 1.40 g/BM carbohydrate, 0.40 g/BM protein, and 0.58 g/BM fat. Both meals were planned by a sports nutritionist.

### 2.8. Muscle Damage, Hormones, Inflammatory and Oxidative Stress Biomarkers

#### 2.8.1. Creatine Kinase (CK)

CK catalyzes the phosphorylation of adenosine diphosphate by creatine phosphate, yielding creatine and adenosine triphosphate. The catalytic concentration is determined using the coupled reactions of hexokinase and glucose-6-phosphate dehydrogenase from the rate of formation of reduced nicotinamide adenine dinucleotide phosphate, measured at 340 nm (Spectrophotometer Stat fax 1904 Plus, Awareness Technology, Inc., Palm City, Florida, USA). The methodology for CK measurements is described in the manual of BioSystems S.A. (Costa Brava, 30. 08030, Barcelona (Spain)).

#### 2.8.2. Hormones

Serum C and T were measured using an Immulite 2000 immunoassay system (Medical Systems, Genoa, Italy), where the intra- and inter-assay coefficient of variance for T were 3.2% and 5.1% and for C 4.2% and 6.9%. Chemil immunoassay technology was used to quantitatively measure in vitro hGH, using the LIAISON^®^ analyzer (DiaSorin S.p.A., Gerenzano, Italy). The coefficient of variance for GH in intra- and inter-assay was 7.1% and 8.2%, respectively. The ratio of T and C (RT/C) was also calculated, in order to analyze the balance between anabolic and catabolic processes.

#### 2.8.3. Inflammatory and Myostatin (IL-6, TNF-α and MYO)

Measurements of highly sensitive-IL-6 (hsIL-6), TNF-α and MYO were performed using high-quality human ELISA kits (eBioscience, Inc., San Diego, CA, USA and Promokine, PromoCell GmbH, Heidelberg, Germany) in accordance with the manufacturer’s recommendations: hs-IL-6 antigen (cat. no. BMS213HS), TNF-α (cat. no. 88734622) and MYO antigen (PK-EL-K1012). Markers’ levels were determined using SpectraMAX^®^ i3x multi-mode microplate reader (Molecular Devices, San Jose, CA, USA). Proinflammatory cytokines and MYO concentrations were quantified using a specific standard curve for each marker. The lower limits of detection for hs-IL-6, TNF-α, and MYO were 0.03 pg/mL, 4 pg/mL, and 0.37 ng/mL, respectively.

#### 2.8.4. Protein Carbonylation and Thiobarbituric Acid Measurements

The carbonyl formation of plasma proteins was analyzed by gel electrophoresis plus Western blot analysis (OxyBlot Protein Oxidation Detection Kit, Millipore). After derivatization with 2,4-dinitrophenylhydrazine, SDS-PAGE gels were transferred to a polyvinylidene difluoride (PVDF) membrane. The membranes were subsequently blocked, washed, and incubated overnight at 4 °C for immunoblotting with an anti-DNP (dinitrophenylhydrazone) primary antibody, according to the kit procedure. The blots were then washed with phosphate-buffered saline and 0.05% Tween 20 and incubated with the secondary goat anti-rabbit IgG/HRP conjugate antibody. After three washes with PBST, the signal was detected with an ECL kit (Pierce ECL detection kit, Thermo Fisher Scientific Inc., Rockford, IL, USA). The emitted chemiluminescent signals were detected using a digital imaging system (ImageQuant LAS500, ThermoFisher, USA) and quantified by ImageJ software (version 1.53). All the samples were loaded with equal amounts of protein, and controls were run with experiments.

Lipid peroxidation in plasma was measured using the Thiobarbituric Acid Reactive Substances (TBARS) assay kit (R&D Systems, Inc., Minneapolis, MN, USA), following the manufacturer’s instructions. Complexes exhibit colors that were determined at 532 mm using SpectraMAX^®^ i3x multi-mode microplate reader (Molecular Devices, San Jose, CA, USA). TBARS were expressed as μM of malondialdehyde (MDA).

### 2.9. Statistical Analysis

IBM Social Sciences software (SPSS, v.21.0, Chicago, IL, USA) was used for statistical analysis. Data are presented as mean ± SD. Homogeneity and normality of the data were checked by Levene and Shapiro–Wilk tests, respectively. For CK and C, repeated-measures two-way ANOVA (>30 subjects) with time factor (five times) and group factor (three groups) was applied. Tukey’s post hoc analysis was performed when significance was found in the ANOVA models. Partial eta squared (ηp2) was calculated as the effect size for the time, group, and time × group interaction of all variables in the ANOVA analysis. Partial eta square thresholds were used as follows: <0.01, irrelevant; ≥0.01, small; ≥0.059, moderate; ≥0.138, large [34]. For T, GH, C, RT/C, IL-6, TNF-α, MYO, and TBARS, the Friedman test (no-parametric) was applied. Durbin–Conover pairwise comparison within the Friedman test. Correlations were performed using Pearson or Spearman. The significance level was set at *p* ≤ 0.05.

## 3. Results

The general characteristics of the participants were as follows: HRC (*n* = 12; age: 23.3 ± 6.6 years; height: 172.9 ± 5.5 cm; weight: 70.9 ± 8.8 kg: BMI 23.7 ± 1.2 kg·m^−2^). HIIT (n = 10; age: 26.2 ± 6.3 years; height: 177.9 ± 7.2 cm; weight: 78.0 ± 8.3 kg: BMI 24.9 ± 0.5 kg·m^−2^) and CT (n = 11; age: 23.3 ± 4.4 years; height: 174.1 ± 4.2 cm; weight: 69.9 ± 7.7 kg: BMI 22.8 ± 1.2 kg·m^−2^).

The effects of the distinct types of acute training on CK (dependent variables) are depicted in Table 1. Significant differences were observed in CK over time (*p* = 0.001; η^2^ = 0.572), and there was a group × time interaction (*p* = 0.021; η^2^ = 0.343) (Figure 1). In addition, a positive correlation was found between ∆B-P0 CK and ∆B-P0 TNF-α in CT (r = 0.770; *p* = 0.006) (Table 2).

On the other hand, the Friedman test detected a group × time interaction (*p* = 0.004) in T (Table 3). Pairwise comparisons with Durbin–Conover showed significant differences in T between HRC-P1 and HIIT-P1 (*p* = 0.006), HRC-P48 and HIIT-P48 (*p* = 0.021), and between CT-P1 and HIIT-P1 (*p* = 0.008) as shown in Figure 1. In addition, a positive correlation was found between ∆B-P0 T and ∆B-P0 MYO in CT (r = 0.998; *p* = 0.035) and an inverse correlation between ∆B-P1 T and ∆B-P1 C in CT (r = −0.814; *p* = 0.004) (Table 2).

Significant time differences were also found in C (*p* = 0.016; η^2^ = 0.443), and there was a group × time interaction (*p* = 0.034; η^2^ = 0.322) (Figure 1). In addition, a positive correlation was found between ∆B-P0 C and ∆B-P0 IL-6 in HIIT (r = 0.767; *p* = 0.016), ∆B-P1 C and ∆B-P1 TNF-α in HIIT (r = 0.702; *p* = 0.035), ∆B-P1 C and ∆B-P1 GH in HRC (r = 0.583; *p* = 0.060), and an inverse correlation between ∆B-P1 C and ∆B-P1 MYO in HRC (r = 0.702; *p* = 0.035). At 24 h post-training session, a significant positive correlation was observed in ∆B-P24 C and ∆B-P24 TNF-α (r = 0.834; *p* = 0.010) in HIIT, ∆B-P24 C and ∆B-P24 TNF-α (r = 0.694; *p* = 0. 018) in HRC, and a significant negative correlation between ∆B-P24_C and ∆B-P24 GH (r = −0.787; *p* = 0.012) in HIIT, and trend between ∆B-P24 C and ∆B-P24 GH (r = −0.599; *p* = 0.052) and in ∆B-P24 C and ∆B-P24 MYO (r = −0.566; *p* = 0.076) in HRC. At 48 h post-training, a significant inverse correlation was found between ∆B-P48 C and ∆B-P48 RT/C (r = −0.980; *p* = 0. 001) in HIIT, between ∆B-P48 C and ∆B-P48 GH (r = −0.614; *p* = 0.045), as well as ∆B-P48 C and ∆B-P48 RT/C (r = −0.896; *p* = <0.01) in CT and between ∆B-P48 C and ∆B-P48 RT/C (r = −0.896; *p* = <0.01) in HRC (Table 2).

Table 4 and Figure 2 show significant differences in hs-IL-6 levels between baseline and P0 among the three groups (*p* < 0.05). Contrary to hs-IL-6, TNF-α levels increased only at P0 with CT (*p* = 0.05) (Table 4). In addition, both inflammatory cytokines recovered baseline levels after 1 h (*p* > 0.05), although a trend was still found for hs-IL-6 levels in CT (*p* = 0.056). In GH, hs-IL-6, and TBARS, we detected significant (Friedman test) group × time interaction (*p* < 0.001), and in MYO a trend in the group × time interaction (*p* = 0.056). In the Durbin–Conover pairwise comparison, significant differences in MYO were found when comparing CT and HRC in P1 (*p* = 0.003) and a trend when comparing HIIT and HRC in P1 (*p* = 0.056) (Table 4). In addition, the Durbin–Conover test detected significant differences in TBARS when comparing HIIT with TC (*p* = 0.012) and HRC (*p* = 0.012) at P1, when comparing HIIT and CT at P24 (*p* = 0.012) and CT and HRC at P24 (*p* = 0.045) (Table 4). In addition, a significant inverse correlation was found between ∆B-P24 GH vs. B-P24 TNF-α in HRC (r = −0.648; *p* = 0.031 (Table 2)). Figure 3 shows plasma Oxyblot results, which revealed no significant time effect (*p* > 0.05) after CT or HRC training. Nonetheless, a significant increase in levels of carbonylated proteins was detected after HIIT exercise and was maintained up to 1 h, when compared to pre-treatment.

## 4. Discussion

Several reports have shown that hormonal factors, coupled with mechanical stimulation of intramuscular signaling, promote a sequence of molecular processes that lead to the adaptive response of the muscle. The initial objective of this study was to evaluate changes in hormonal, inflammatory muscle damage-atrophy, and oxidative stress changes in both the acute and recovery phases after CT, HRC, and HIIT training. This research found that levels of CK, T, C, GH, hs-IL-6, and TBARS were significantly impaired, with a trend in MYO, after a single training session of CT, HRC, and HIIT with substantial differences between each training protocol in recreationally active subjects. Specifically, HRC training generates higher T levels during the first 30 min (P1) until 48 h (P48) after the end of the training session compared to HIIT. In addition, HRC training generates lower MYO levels at P1 compared to HIIT and CT post-exercise. These results provide further support for the hypothesis that HRC training induces an optimal biochemical-hormonal environment that would promote protein synthesis (anabolism) and muscle tissue repair and growth.

### 4.1. Testosterone

To our knowledge, this is the first study to observe that HRC training produced an increase in T compared to HIIT after the end of the exercise (P1–P48). In order to understand the T response to different types of training, Kraemer et al. [35] investigated the effect of altering intensity while maintaining a constant total workload and found that when intensity was decreased, there was a lower T response. However, when the number of repetitions was maintained constant, higher intensity and volume increased the T level. On the other hand, Raastad et al. [36] demonstrated that 3 sets of 6 repetitions of three lower body exercises at 100% of 6RM significantly increased T, but not at 70–76% of 6RM. Likewise, 5 sets of 10 repetitions at 10RM significantly increased T, whereas 70% or 40% of 10RM did not modify T levels. In addition, the volume of training is an important factor in the modulation of the hormonal system, since it has been shown that 6 sets, but not 1 set, of 10-rep squats increased T following exercise [37]. Another determinant for an increase in T following endurance exercise is the muscle mass involved. Even when exercised vigorously, the involvement of small muscle groups does not increase T levels above resting concentrations [38]. Exercises, such as the squat [37,39] and Olympic lifts [40], that use large muscle mass, generate a greater elevation in T compared to exercises with less muscle mass [41,42]. Moreover, the recovery period between sets can influence post-exercise T stimulation. It has been shown that loads above 10RM with a high total workload and a short rest (1 min) produced a significantly higher increase in T compared to a longer rest (3 min) [35].

Mechanistically, these findings suggest that testosterone secretion is closely tied to training variables that modulate neuromuscular and metabolic stress. The combination of high-load HRC with a short recovery between sets (45 s) seems to be an optimal configuration to generate a greater increase in post-exercise T compared to CT and HIIT. This may be due to the fact that HRC can generate higher metabolic stress influenced by a high energy demand with a high lactate production [35] and oxygen consumption [43] due to a short recovery (45 s) between sets and the high loads used (6RM) [36]. This metabolic environment—characterized by high lactate and oxygen demand—can amplify hypothalamic-pituitary-gonadal axis activation, thereby promoting a more robust release of testosterone.

In contrast, there is also scientific evidence that establishes that the acute increase in T after resistance exercise does not influence protein synthesis and hypertrophy in the long term [44]. This narrative review shows how decreased T is related to decreased muscle strength and function, and how the administration of high concentrations of exogenous T promotes muscle hypertrophy in both women and men. In addition, T can act through non-genomic (cell signaling) and intracrine action, the latter referring to how local T precursors (dihydrotestosterone, etc.) can be converted to T within skeletal muscle. [45].

T is a strong androgenic-anabolic effect hormone, as it regulates muscle growth by increasing protein synthesis and inhibiting protein degradation [46]. The anabolic effects of T are induced by its binding to the intracellular androgen receptor (AR), resulting in translocation to the nucleus where the AR-T complex promotes the transcription of specific genes [47]. The anti-catabolic effects of T stem come primarily via inhibition of the C signaling cascade through the blockade of the glucocorticoid receptor [48]. Conversely, excess glucocorticoids can interfere with T signaling and decrease T production in Leydig cells [46]. At the muscular level, T increases protein synthesis [49] and intramuscular uptake of amino acids [50], leading to an improved net protein balance [51]. At the molecular level, T administration has been shown to increase glucose transport through increased expression of glucose transporter type 4 (GLUT4), increased insulin signaling through increased expression of insulin receptor substrate 1 and 2 (IRS1 and IRS2), and increased glycogen synthesis through increased glycogen synthase activity in skeletal muscle [52]. In addition, it can increase lactate transport through regulation of the expression of monocarboxylate transporter 1 (MCT1) and 4 (MCT4) proteins [53], and increase activation, proliferation, mobilization, differentiation and incorporation of satellite cells in skeletal muscle [54]. Moreover, T can increase GH [55] and activin receptor-like kinase 4/5 (Akt 4/5) mammalian target of rapamycin (mTOR) pathway activation [56]. mTOR is a very important intracellular signaling intermediary that promotes cell growth by modulating anabolic processes such as protein and lipid synthesis [57].

The process of muscle recovery after endurance exercise and hypertrophy comprises a complex interconnected system, where internal (genetics, epigenetics, intracellular signaling, etc.) and external factors (nutrition, rest, type of training, etc.) intervene. There is ambiguity about the effect of T concentrations after endurance exercise on hypertrophy, there is scientific evidence that T intervenes by various mechanisms in the recovery process after endurance exercise.

### 4.2. Growth Hormone

In relation to GH, statistical analysis (Friedman test) detected significant differences between groups, but there was no significant difference in the pairwise comparison. It is known that GH is mainly stimulated by exercise with a high anaerobic component (↓pH and ↑H^+^) [35,58]. Resistance training characterized by short rest periods between sets and exercises leads to higher GH blood levels during recovery in both men and women [29]. However, when rest is longer, the GH response in recovery is significantly lower [35,59]. In addition, total workload is important for GH levels, as the elevation of GH after strength exercise increases with workload [35]. T may also modulate GH concentrations, since T administration reduced protein oxidation in prepubertal GH-deficient boys, as measured by leucine oxidation, but did not alter measures of protein synthesis. However, when GH and T were administered together, there was an increase in protein synthesis [60]. This indicates that minimal concentrations of GH may be required for the actions of T, with a synergistic effect on protein synthesis. At the molecular level, GH can acutely trigger protein synthesis through the activation of mTOR and specifically through the rapamycin-sensitive mTOR complex 1 (mTORC1) [61].

### 4.3. Myostatin

A key factor involved in protein balance is MYO [62]. In our study, we found a group × time interaction in MYO, and we found a trend when comparing HRC vs. HIIT at P0. Raue et al. [63] observed that an acute resistance exercise session decreased MYO mRNA 2.2-fold in biopsies from young and older women taken 4 h after a single session of 3 sets of 10 repetitions at 70% of 1RM knee extension, which was also following biopsies taken after 24 h of knee extensor resistance exercise [64]. Conversely, Willoughby [65] found a significant increase in MYO mRNA and protein in vastus muscle and blood samples collected 15 min after the last session of a lower extremity resistance training program (6 or 12 weeks with three weekly sessions of three sets of six to eight repetitions of 85–90% of 1RM of leg press and knee extension). Therefore, the relationship between resistance training and MYO modulation requires further investigation. Furthermore, it should be clarified that an acute decrease is not synonymous with a chronic decrease in myostatin. In a cluster analysis study in resistance-trained humans, MYO mRNA levels decreased significantly in a single knee extension exercise 24 h after completing the last session of a 16-week training program [66]. The decrease in MYO was similar for non-responders, modest responders, and extreme responders with respect to changes in muscle fiber size [66]. Herein, these data indicate that the decrease in MYO expression was not associated with increased muscle growth in response to resistance training. At the molecular level, MYO inhibits the activation of the Akt/mTOR/p70S6 protein synthesis pathway, which mediates both myoblast differentiation and myotube hypertrophy [67]. In the current study, we extend previous findings by demonstrating that higher levels of T and lower levels of MYO after a single HRC exercise session generate a hormonal profile favorable for increased protein synthesis and muscle mass compared to HIIT and CT. In addition, inhibition of myostatin signaling in P0 after HIIT could result in significant muscle hypertrophy together with higher body glucose metabolism [68], suggesting distinct activation mechanisms compared to endurance exercise.

### 4.4. Inflammatory Response

In the case of inflammatory markers, we found a group × time interaction in IL-6 and TNF-α, but without any significance in the pairwise comparison. A possible explanation for this might be that the IL-6 generated in skeletal muscle and satellite cells is increased mainly in overloaded muscles, suggesting a role in cell proliferation, myonuclear enlargement, and hypertrophy [12]. These results are in line with reactive protein thiol levels, suggesting a transient pro-oxidative state after acute exercise sessions. IL-6 acts not only as a proinflammatory cytokine but also as a myokine, contributing to muscle remodeling and regeneration processes. Supporting this dual role, a recent study by Peake et al. [12] highlighted that IL-6 releases following resistance training are modulated by both exercise intensity and muscle damage, serving as a signal for satellite cell recruitment and tissue repair. Additionally, the lack of pairwise significance may reflect the complex time course of cytokine release, which can vary considerably depending on training history and individual recovery capacity. Indeed, other investigations have reported that trained individuals exhibit a blunted inflammatory response to acute exercise, likely due to adaptations in cytokine signaling and immuno-endocrine regulation developed through regular training [12,69].

### 4.5. Oxidative Stress

Another important finding was that significant differences between groups were observed in TBARS (a marker of lipid peroxidation) both in the group × time interaction and in the pairwise comparison. The most striking difference was found s when comparing HRC vs. HIIT and HIIT vs. CT at P1, HIIT vs. CT, and HRC vs. CT at P24, suggesting a longer pro-oxidative state after HRC. Deminice et al. [70] found that a traditional interval resistance training session (3 × 10 repetitions at 75% of 1RM, with 90 s of passive rest) increased TBARS by 40%, with no change in a circuit training session (3 × 10 repetitions at 75% of 1RM, alternating 2 exercises with different muscle groups). McBride et al. [71] observed that in trained young men who performed resistance training (3 sets of 10 repetitions at 1RM for 8 exercises) lipoperoxidation increased in blood as well. It seems then possible that high-intensity resistance exercise increases free radical production. In contrast, no significant change in MDA was observed after performing six 10 s sprints or barbell squats (equal to the amount of work performed) in trained individuals [72]. On the other hand, another study analyzed the effect of oxidative stress after performing 4 sets of the squat exercise using either a low-intensity, high-volume (15 repetitions at 60% 1 repetition maximum 1RM) or high-intensity, low-volume (4 repetitions at 90% 1RM load), and found no significant difference between groups in plasma MDA after exercise in trained subjects [73]. It is then difficult to clarify which components of resistance exercise have the greatest impact on oxidative stress [74]. It should be noted that the increase in oxidative markers is biphasic, with a peak at 24 and 72 h after exercise. The former increase is probably caused by the repeated ischemia–reperfusion state after exercise, while the latter depends on inflammation at the damaged site [75,76]. Other studies demonstrate that oxidative stress is dependent on the training status, with DNA damage less evident in trained athletes [76]. This intriguing result may reflect an interference between oxidative stress and recovering inflammation from injury in the skeletal muscle.

A 120% and 68% increase in TBARS has been reported after high and moderate intensity running, respectively [77]. The higher TBARS levels found after HRC may be due to the binding of the high-intensity component to the short recovery component, generating higher metabolic and oxidative stress, compared to CT and HIIT. Since our exercise sessions increased lipid peroxidation (especially after HRC), we investigated the protein carbonyl levels, another marker of oxidative stress. In accordance with the above data, protein carbonyl levels were transiently higher in 0 h (Baseline vs. P0) and 1 h groups (Baseline vs. P1) after HIIT, then returning to baseline levels. However, no differences at P0 nor P1 were observed for CT and HRC. Therefore, our results show that a single high-intensity exercise induces an acute oxidative status, which might help to modulate a physiological defense to low levels of ROS that are implicated in exercise-induced skeletal muscle fatigue.

In addition, the time duration of endurance training should be considered when endurance and strength are performed in the same training session. HRC produces higher levels of TBARS at p0 and P1 in HRC and HIIT compared to CT and T post-exercise (48 h) compared to HIIT, which might suggest that this oxidative stress is related to a better hormonal environment. It has been shown that myocytes contain a myriad of antioxidant mechanisms to reduce oxidative damage during periods of increased ROS production, and myostatin inhibition enhances the erythrocyte antioxidant capacity [78], which was accordingly found in P0 and P1 after HRC training.

This research provides new insights into the acute and delayed recovery phase effects of different types of resisted training (HIIT, CT, and HRC) on hormonal, inflammatory, and muscle damage biomarkers in recreationally active individuals. Our results show that HRC generates a better anabolic environment after a training session than HIIT and CT, which may contribute to optimal recovery. These three types of resisted training are commonly used by many trainers worldwide to generate the physiological-metabolic adaptations needed to improve their performance. Therefore, our findings allow us to establish that HRC does not generate superior muscle damage to HIIT and CT, with the advantage of generating a positive anabolic environment that would favor rapid regeneration of damaged fibers after HRC training. These beneficial effects of HRC training with respect to other types of training, is that at times of the sports season when there is a lot of physiological and mechanical stress could also reduce the risk of injury.

## 5. Conclusions

These experiments confirmed that acute high-intensity resistance circuit training induces lower levels of MYO in the acute recovery phase (P1) compared to high-intensity interval training and concurrent training and higher levels of T in the acute (P1) and late recovery phase (P48) compared to HIIT from a single training session. Therefore, the high-intensity circuit is a resistance training method that promotes an optimal hormonal environment for protein synthesis and hypertrophy post-exercise after a single training session. The insights gained from this study may be of assistance to optimize hormonal responses in order to produce the most beneficial biological changes.

## Figures and Tables

**Figure 1 sports-13-00184-f001:**
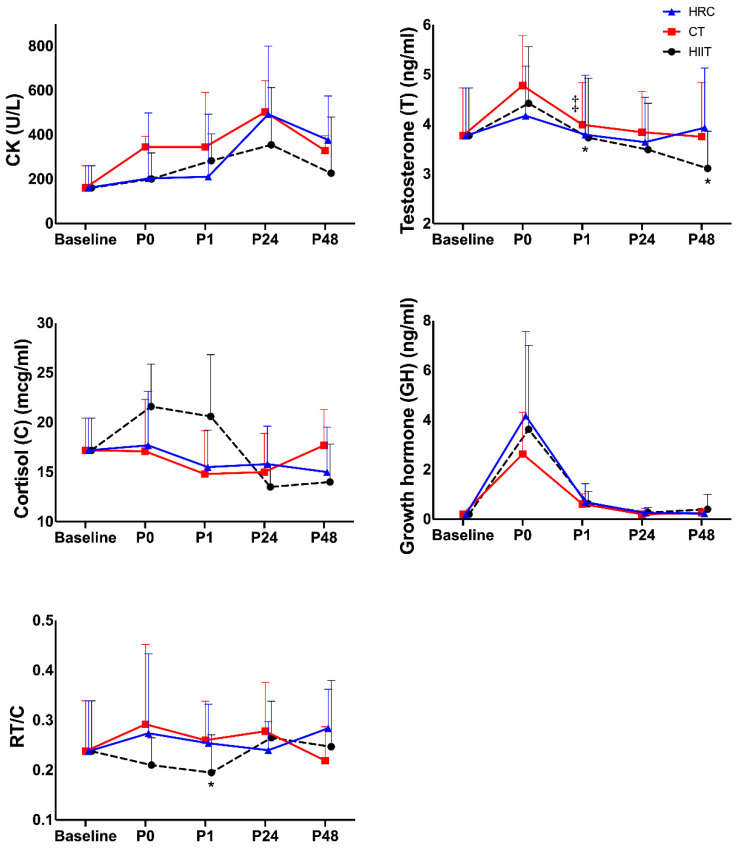
Time course of CK, T, C, GH, and RT/C from baseline to 48 h post-training. * = significant differences between HRC and HIIT, *p* = <0.05. ‡ = significant differences between HIIT and CT, *p*-values = ≤0.05.

**Figure 2 sports-13-00184-f002:**
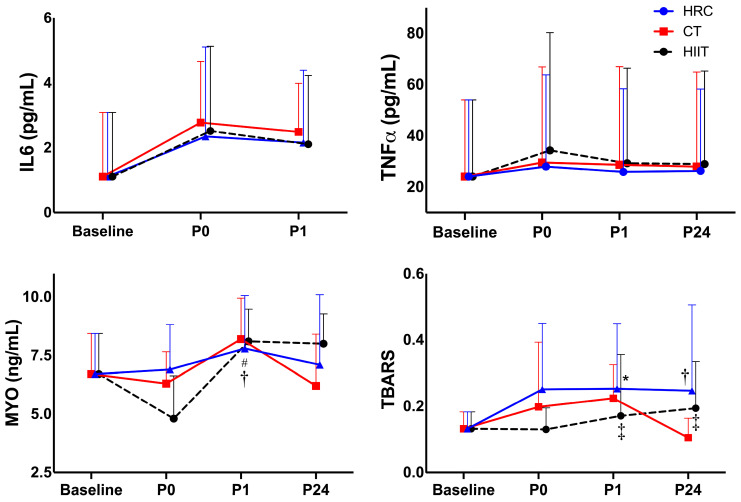
Time course of IL-6, TNF-α, MYO, and TBARS from baseline to 24 h post-training. * = significant differences between HRC and HIIT, *p*-values = ≤0.05. † = significant differences between HRC and CT, *p*-values = ≤0.05. ‡ = significant differences between HIIT and CT, *p*-values = ≤0.05. # = trend between HRC and HIIT, *p*-values = 0.05–0.06.

**Figure 3 sports-13-00184-f003:**
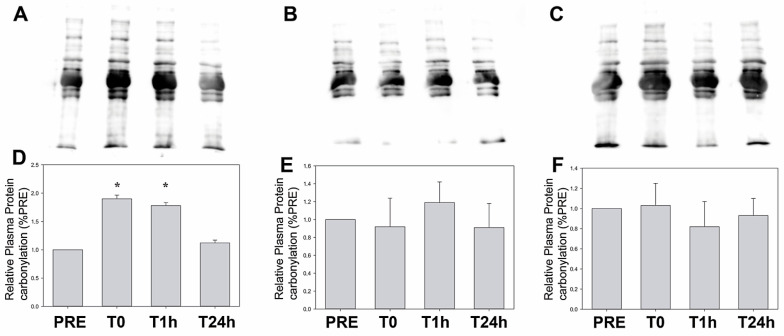
Levels of carbonylated proteins, as assessed by the oxyblot technique. Plots of the protein carbonyl content are shown (**A**) HIIT; (**B**) CT and (**C**) HRC. Mean densitometry data are shown in the bottom graphs (**D**) HIIT; (**E**) CT and (**F**) HRC. Pre: pre-treatment (baseline); T0: immediately after exercise (P0); T1h: 1 h after exercise (P1); T24h: 24 h after exercise (P24). Data were expressed as relative to pre-treatment and * = indicates a significant statistical difference (*p* < 0.05) compared to pre-treatment.

**Table 1 sports-13-00184-t001:** Results of the two-way ANOVA and Friedman analysis of the variables analyzed in this study during the three types of training (HIIT, CT, and HRC) at different blood collection points.

		Baseline	P0	P1	P24	P48	Time *p*-Value	Group *p*-Value	Interaction Time × Group *p*-Value
**CK** (UI/L)	HIIT	161 (99.4)	203 (117.0)	211 (121.0)	493 (258)	377 (253.0)	**0.001**	0.435	**0.021**
	CT	161 (99.4)	201 (48.5)	283 (246.0)	355 (1410)	227 (67.1)			
	HRC	161 (99.4)	251 (203.0)	345 (282.0)	394 (241.0)	329 (198.0)			
	η^2^						0.572	0.153	0.343

In bold are *p*-values = ≤0.05 and trends between 0.05 and 0.07. CK: creatin kinase.

**Table 2 sports-13-00184-t002:** Correlations found in the different types of training comparing the different measurement points of all the biochemical variables analyzed.

Correlations	r	*p*-Value
∆B-P0_CT_CK vs. B-P0_CT_TNF-α	0.770	0.006
∆B-P0_HRC_CK vs. B-P0_HRC_C	0.626	0.039
∆B-P0_CT_T vs. B-P0_CT_RT/C	0.818	0.004
∆B-P0_CT_T vs. B-P0_CT_MYO	0.998	0.035
∆B-P0_HRC_T vs. B-P0_HRC_GH	0.570	0.067
∆B-P0_HIIT_C vs. B-P0_HIIT_IL-6	0.767	0.016
∆B-P0_CT_C vs. B-P0_CT_RT/C	−0.782	0.008
∆B-P0_HRC_C vs. B-P0_HRC_RT/C	−0.741	0.014
∆B-P1_HIIT_CK vs. B-P1_HIIT_TBARS	−0.664	0.051
∆B-P1_CT_CK vs. B-P1_CT_IL-6	−0.722	0.028
∆B-P1_CT_T vs. B-P1_CT_C	−0.814	0.004
∆B-P1_CT_T vs. B-P1_CT_RT/C	0.682	0.043
∆B-P1_HIIT_C vs. B-P1_HIIT_RT/C	−0.667	0.035
∆B-P1_HIIT_C vs. B-P1_HIIT_IL-6	−0.683	0.029
∆B-P1_HIIT_C vs. B-P1_HIIT_TNF-α	0.702	0.035
∆B-P1_HIIT_C vs. B-P1_HIIT_MYO	−0.679	0.044
∆B-P1_CT_C vs. B-P1_CT_RT/C	−0.957	<0.001
∆B-P1_HRC_C vs. B-P1_HRC_GH	0.583	0.060
∆B-P1_HRC_C vs. B-P1_HRC_RT/C	−0.742	0.014
∆B-P1_HRC_C vs. B-P1_HRC_MYO	−0.727	0.011
∆B-P24_HIIT_CK vs. B-P24_HIIT_C	−0.781	0.013
∆B-P24_HIIT_T vs. B-P24_HIIT_RT/C	0.885	0.002
∆B-P24_HIIT_C vs. B-P24_HIIT_GH	−0.787	0.012
∆B-P24_HIIT_C vs. B-P24_HIIT_RT/C	−0.754	0.019
∆B-P24_HIIT_C vs. B-P24_HIIT_TNF-α	0.834	0.010
∆B-P24_HRC_C vs. B-P24_HRC_GH	−0.599	0.052
∆B-P24_HRC_C vs. B-P24_HRC_RT/C	−0.911	<0.01
∆B-P24_HRC_C vs. B-P24_HRC_TNF-α	0.694	0.018
∆B-P24_HRC_C vs. B-P24_HRC_MYO	−0.566	0.076
∆B-P24_HRC_GH vs. B-P24_HRC_TNF	−0.648	0.031
∆B-P48_HIIT_C vs. B-P48_HIIT_RT/C	−0.980	0.001
∆B-P48_CT_C vs. B-P48_CT_GH	−0.614	0.045
∆B-P48_CT_C vs. B-P48_CT_RT/C	−0.896	<0.01
∆B-P48_HRC_C vs. B-P48_HRC_RT/C	−0.930	<0.01

B: baseline; P0: post-training; P1: 1 h post-training; P24: 24 h post-training; P48: 48 h post-training; CK: creatin kinase; T: testosterone; C: cortisol; GH: growth hormone; RT/C: testosterone/cortisol ratio; IL-6: interleukin 6; TNF-α: tumoral necrosis factor α; MYO: myostatin; TBARS: thiobarbituric acid reactive substances.

**Table 3 sports-13-00184-t003:** Results of the two-way ANOVA and Friedman analysis of testosterone, cortisol, growth hormone, and testosterone/cortisol ratio during the three types of training (HIIT, CT, and HRC) at different blood collection points.

		Baseline	P0	P1	P24	P48	Time *p*-Value	Group *p*-Value	Interaction Time × Group *p*-Value
**T** (ng/mL)	HIIT	3.77 (0.96)	4.42 (1.14)	3.73 (1.20)	3.49 (0.93)	3.11 (0.75)			**0.004**
	CT	3.77 (0.96)	4.78 (1.00)	3.99 ‡ (0.85)	3.84 (0.82)	3.75 (1.09)			
	HRC	3.77 (0.96)	4.17 (1.02)	3.79 (1.25) *	3.64 (0.92)	3.93 (1.19) *			
	η^2^								
**C** (mcg/dL)	HIIT	17.2 (3.22)	21.6 (4.27)	20.6 (6.23)	13.5 (2.24)	14.0 (3.83)	**0.016**	0.429	**0.034**
	CT	17.2 (3.22)	17.1 (5.22)	14.8 (4.35)	15.0 (3.88)	17.7 (3.60)			
	HRC	17.2 (3.22)	17.7 (5.44)	15.5 (3.72)	15.8 (3.82)	15.0 (4.53)			
	η^2^						0.443	0.156	0.322
**GH** (ng/mL)	HIIT	0.20 (0.00)	3.62 (3.38)	0.62 (0.50)	0.27 (0.20)	0.40 (0.60)			**<0.001**
	CT	0.20 (0.00)	2.63 (1.68)	0.61 (0.41)	0.20 (0.00)	0.26 (0.18)			
	HRC	0.20 (0.00)	4.16 (3.41)	0.69 (0.74)	0.26 (0.18)	0.23 (0.09)			
	η^2^								
**RT/C**	HIIT	0.237 (0.10)	0.210 (0.06)	0.195 (0.08)	0.267 (0.07)	0.247 (0.13)			0.188
	CT	0.237 (0.10)	0.292 (0.16)	0.260 (0.08)	0.279 (0.10)	0.219 (0.07)			
	HRC	0.237 (0.10)	0.274 (0.16)	0.254 (0.08) *	0.239 (0.06)	0.284 (0.08)			
	η^2^								

In bold are *p*-values = ≤0.05 and trends between 0.05 and 0.07. T: testosterone; C: cortisol; GH: growth hormone; RT/C: testosterone/cortisol ratio. * = significant differences between HRC and HIIT, *p*-values = ≤0.05, ‡ = significant differences between HIIT and CT.

**Table 4 sports-13-00184-t004:** Results of the two-way ANOVA and Friedman analysis of inflammatory markers, myostatin, and TBARS during the three types of training (HIIT, CT and HRC) at different blood collection points.

		Baseline	P0	P1	P24	Time *p*-Value	Group *p*-Value	Interaction Time × Group *p*-Value
**hs-IL-6** (pg/mL)	HIIT	1.11 (1.98)	2.52 (2.61)	2.10 (2.12)				**<0.001**
	CT	1.11 (1.98)	2.78 (2.19)	2.49 (1.50)				
	HRC	1.11 (1.98)	2.35 (2.76)	2.16 (2.23)				
	η^2^							
**TNF-α** (pg/mL)	HIIT	24.1 (29.9)	34.3 (46.0)	29.3 (37.1)	29.0 (36.3)			0.158
	CT	24.1 (29.9)	29.6 (37.3)	25.9 (32.5)	28.0 (36.9)			
	HRC	24.1 (29.9)	28.0 (35.8)	25.9 (32.5)	26.3 (32.0)			
	η^2^							
**MYO** (ng/mL)	HIIT	6.66 (1.74)	4.81 (1.83)	8.13 (1.38)	8.03 (1.28)			**0.056**
	CT	6.66 (1.74)	6.29 (1.36)	8.25 (1.75)	6.17 (2.21)			
	HRC	6.66 (1.74)	6.93 (1.92)	7.83 (2.26) †#	7.06 (3.00)			
	η^2^							
**TBARS** (µM)	HIIT	0.132 (0.05)	0.130 (0.07) ‡	0.171 (0.19) ‡	0.194 (0.14)			**<0.001**
	CT	0.132 (0.05)	0.198 (0.20)	0.224 (0.10)	0.105 (0.06)			
	HRC	0.132 (0.05)	0.251 (0.20) *	0.253 (0.20) †	0.247 (0.26)			
	η^2^							

In bold are *p*-values = ≤0.05 and trends between 0.05 and 0.07. hs-IL-6: high sensitive- interleukin-6; TNF-α: tumoral necrosis factor α; MYO: myostatin; TBARS: thiobarbituric acid reactive substances. * = significant differences between HRC and HIIT, *p*-values = ≤0.05. † = significant differences between HRC and CT, *p*-values = ≤0.05. ‡ = significant differences between HIIT and CT, *p*-values = ≤0.05. # = trend between HRC and HIIT, *p*-values = 0.05–0.06.

## Data Availability

The data that support the findings of this study are available from the corresponding author upon reasonable request.

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
