# Peer review of "Comparison of Hormonal, Inflammatory, Muscle Damage and Oxidative Stress Biomarkers Changes in Response to High-Intensity Interval, Circuit and Concurrent Exercise Bouts"

_sports, 2025, doi:10.3390/sports13060184_

Round 1

Reviewer 1 Report

Comments and Suggestions for Authors

The submitted manuscript aimed to examine the acute and recovery-phase biomarker responses (hormonal, inflammatory, oxidative stress, and muscle damage) to three types of resistance-based training: HIIT, CT, and HRC. Based on their findings, the authors report that HRC generates higher levels of testosterone than HIIT in the acute (P1) and late (P48) phases of recovery and produces lower levels of myostatin than CT and HIIT (P1) in the acute phase of recovery. The topic is valuable and always of interest, especially in an applied exercise and sports context. Please find next my comments that may be of help.

  • The title and abstract refer to the “time course” of biomarker responses, yet much of the discussion fixates on comparing types of training. These are related but conceptually distinct Are the authors testing whether different protocols affect biomarkers differently or whether they follow similar temporal dynamics? Right now, it’s both — and neither is handled appropriately I think.
  • The authors state that this is a crossover study in their methodology. Is it? Because in the results section they state that HRC (n = 12), HIIT (n = 10) and CT (n = 11). If it is a parallel-group study, is the sample size sufficient to ensure statistical power?
  • “Recreationally active” is a catch-all Please provide some information.
  • The results section is bloated with an excessive number of biomarkers and post hoc correlations. Many seem exploratory without correction for multiple comparisons (major issue – see next). Tables are hard to digest and lack clarity. For instance, repetitive listing of same baseline values in different groups is hard to follow.
  • The authors bounce between ANOVA, Friedman, and dozens of Pearson/Spearman correlations seemingly at random. No correction for Type I error across ~30+ statistical tests. Please, apply corrections for multiple comparisons (e.g., Bonferroni or FDR) and you could also use heatmaps or summary graphs instead of overwhelming tables. If it is a crossover study, the whole statistical analysis approach should be revisited.
  • The discussion makes some bold and causal physiological claims (e.g., HRC “promotes hypertrophy” or “reduces injury risk”) based solely on acute hormonal changes. There is clearly a choice of studies to support authors’ claims and no discussion of opposing findings (e.g., the debate on post-exercise humoral factors relevance – see PubMedID: 39190607).
  • The manuscript, especially discussion, is overwritten and repetitive. Entire paragraphs are copy-pasted textbook physiology with no connection to results
  • Another major issue is the use of TBARS, which is nowadays considered almost inappropriate as a biomarker to evaluate oxidative stress. Sometimes called as T(hat) B(loody) A(ssay) (see PubMedID: 28371751).
  • Figure 3. Higher resolution in needed.
  • The concept of ‘a trend’ towards significance does not exist in statistics, although it is frequently mentioned by authors. No one knows if and how the results will change if we add observations.
  • The discussion on redox biology could be improved. I mean, with regards to exercise, mitochondria are not the main source of ROS. Instead, NADPH oxidases are the key molecule here.

Author Response

REVIEWER 1

The submitted manuscript aimed to examine the acute and recovery-phase biomarker responses (hormonal, inflammatory, oxidative stress, and muscle damage) to three types of resistance-based training: HIIT, CT, and HRC. Based on their findings, the authors report that HRC generates higher levels of testosterone than HIIT in the acute (P1) and late (P48) phases of recovery and produces lower levels of myostatin than CT and HIIT (P1) in the acute phase of recovery. The topic is valuable and always of interest, especially in an applied exercise and sports context. Please find next my comments that may be of help.

We thank the reviewer for his/her constructive and helpful feedback on our manuscript. We have replied to each comment in the section below and have introduced the corresponding edits into the manuscript using Word’s track changes.

The title and abstract refer to the “time course” of biomarker responses, yet much of the discussion fixates on comparing types of training. These are related but conceptually distinct Are the authors testing whether different protocols affect biomarkers differently or whether they follow similar temporal dynamics? Right now, it’s both — and neither is handled appropriately I think.

Response: We have modified our title to “Comparison of hormonal, inflammatory, muscle damage and oxidative stress biomarker changes in response to high-intensity interval, circuit and concurrent exercise bouts”.

The authors state that this is a crossover study in their methodology. Is it? Because in the results section they state that HRC (n = 12), HIIT (n = 10) and CT (n = 11). If it is a parallel-group study, is the sample size sufficient to ensure statistical power?

Response: The study was a crossover design and not a parallel-group study. The reason why sample sizes are different is because there were participants who were unable to attend the training day.

“Recreationally active” is a catch-all Please provide some information.

Response: Amended.

The results section is bloated with an excessive number of biomarkers and post hoc correlations. Many seem exploratory without correction for multiple comparisons (major issue – see next). Tables are hard to digest and lack clarity. For instance, repetitive listing of same baseline values in different groups is hard to follow.

Response: Thank you for your comment. In the statistical analysis of the methodology section, we explained that we used Tukey's post hoc analysis for the correction of multiple comparisons in ANOVA. Regarding the tables, we have grouped the parameters analyzed in several tables.

The authors bounce between ANOVA, Friedman, and dozens of Pearson/Spearman correlations seemingly at random. No correction for Type I error across ~30+ statistical tests. Please, apply corrections for multiple comparisons (e.g., Bonferroni or FDR) and you could also use heatmaps or summary graphs instead of overwhelming tables. If it is a crossover study, the whole statistical analysis approach should be revisited.

Response: Thank you for your comment. The selection criteria for the statistical method were based on the normality of the sample distribution and the number of data collected for each parameter (N). Therefore, we used ANOVA to analyze variables that did not have a normal distribution but exceeded 30 subjects measured for all measurement points. Furthermore, if there was a significance interaction of the factors analyzed, Tukey's post hoc analysis for the correction of multiple comparisons was applied. The central limit theorem states that with large sample sizes (n= >30), the sampling distribution of the mean tends to be normal, although if the data violate other assumptions this would not justify 100% use of parametric tests. (doi: 10.1146/annurev.publhealth.23.100901.140546.)

For variables that did not have a normal distribution (i.e., N<30), we used the Friedman test, which does not allow us to perform any correction analysis for multiple comparisons. The Friedman test is the most reliable alternative in terms of robustness to repeated-measures ANOVA when the assumptions of normality are not met. (doi.org/10.1201/9780429186196.)

The discussion makes some bold and causal physiological claims (e.g., HRC “promotes hypertrophy” or “reduces injury risk”) based solely on acute hormonal changes. There is clearly a choice of studies to support authors’ claims and no discussion of opposing findings (e.g., the debate on post-exercise humoral factors relevance – see PubMedID: 39190607).

Response: We thank the reviewer for the recommended article. We have added a new paragraph to the discussion section to address ambiguity in the subject, since muscle hypertrophy is driven by a complex system that is affected by internal and external factors.

The manuscript, especially discussion, is overwritten and repetitive. Entire paragraphs are copy-pasted textbook physiology with no connection to results.

Response: Thank you for your comment, but some biochemical and metabolic mechanisms cannot be summarized and sometimes have to be described literally.

Another major issue is the use of TBARS, which is nowadays considered almost inappropriate as a biomarker to evaluate oxidative stress. Sometimes called as T(hat) B(loody) A(ssay) (see PubMedID: 28371751).

Response: We understand your concern, for future studies we will use other biomarkers of oxidative stress.

Figure 3. Higher resolution in needed.

Response: Amended.

The concept of ‘a trend’ towards significance does not exist in statistics, although it is frequently mentioned by authors. No one knows if and how the results will change if we add observations.

Response: Thank you for your comment.  We use the concept “trend” based on Gravetter and Wallnau, who recognized the use of the concept trend in the description of results that are close to the threshold of statistical significance (p≥0.05), but do not rigorously reach it. These authors mention that a result that is not statistically significant does not necessarily mean that there is no effect; it may indicate a trend that did not reach significance due to insufficient sample size or data variability. They also warn that a trend should not be confused with a definitive conclusion and that using the concept “trend” without a statistical basis can lead to misinterpretation or over-interpretation of the data. Gravetter, F. J., & Wallnau, L. B. (2017). Statistics for the Behavioral Sciences (10th ed.). Cengage Learning.

The discussion on redox biology could be improved. I mean, with regards to exercise, mitochondria are not the main source of ROS. Instead, NADPH oxidases are the key molecule here.

Response: Thank you for your valuable comment. We agree that NADPH oxidases, particularly NOX2 and NOX4, have been identified as key contributors to reactive oxygen species (ROS) production in skeletal muscle during exercise, especially in response to metabolic stress.

However, we would like to note that the contribution of mitochondrial ROS cannot be entirely excluded, particularly during intense or prolonged exercise when mitochondrial oxygen consumption is substantially elevated. Recent literature suggests that the relative contribution of ROS sources may vary depending on exercise modality, intensity, and duration. For instance, mitochondrial ROS production has been shown to increase during sustained, high-intensity endurance exercise due to elevated electron flux through the electron transport chain, particularly at complex I and III.

To address this, we have revised the discussion, highlighting NADPH oxidases as major ROS producers during exercise while also acknowledging the context-dependent contribution of mitochondrial sources.

Author comment: We appreciate all the comments made on our manuscript, which helped improve it’s quality.

REVIEWER 1

The submitted manuscript aimed to examine the acute and recovery-phase biomarker responses (hormonal, inflammatory, oxidative stress, and muscle damage) to three types of resistance-based training: HIIT, CT, and HRC. Based on their findings, the authors report that HRC generates higher levels of testosterone than HIIT in the acute (P1) and late (P48) phases of recovery and produces lower levels of myostatin than CT and HIIT (P1) in the acute phase of recovery. The topic is valuable and always of interest, especially in an applied exercise and sports context. Please find next my comments that may be of help.

We thank the reviewer for his/her constructive and helpful feedback on our manuscript. We have replied to each comment in the section below and have introduced the corresponding edits into the manuscript using Word’s track changes.

The title and abstract refer to the “time course” of biomarker responses, yet much of the discussion fixates on comparing types of training. These are related but conceptually distinct Are the authors testing whether different protocols affect biomarkers differently or whether they follow similar temporal dynamics? Right now, it’s both — and neither is handled appropriately I think.

Response: We have modified our title to “Comparison of hormonal, inflammatory, muscle damage and oxidative stress biomarker changes in response to high-intensity interval, circuit and concurrent exercise bouts”.

The authors state that this is a crossover study in their methodology. Is it? Because in the results section they state that HRC (n = 12), HIIT (n = 10) and CT (n = 11). If it is a parallel-group study, is the sample size sufficient to ensure statistical power?

Response: The study was a crossover design and not a parallel-group study. The reason why sample sizes are different is because there were participants who were unable to attend the training day.

“Recreationally active” is a catch-all Please provide some information.

Response: Amended.

The results section is bloated with an excessive number of biomarkers and post hoc correlations. Many seem exploratory without correction for multiple comparisons (major issue – see next). Tables are hard to digest and lack clarity. For instance, repetitive listing of same baseline values in different groups is hard to follow.

Response: Thank you for your comment. In the statistical analysis of the methodology section, we explained that we used Tukey's post hoc analysis for the correction of multiple comparisons in ANOVA. Regarding the tables, we have grouped the parameters analyzed in several tables.

The authors bounce between ANOVA, Friedman, and dozens of Pearson/Spearman correlations seemingly at random. No correction for Type I error across ~30+ statistical tests. Please, apply corrections for multiple comparisons (e.g., Bonferroni or FDR) and you could also use heatmaps or summary graphs instead of overwhelming tables. If it is a crossover study, the whole statistical analysis approach should be revisited.

Response: Thank you for your comment. The selection criteria for the statistical method were based on the normality of the sample distribution and the number of data collected for each parameter (N). Therefore, we used ANOVA to analyze variables that did not have a normal distribution but exceeded 30 subjects measured for all measurement points. Furthermore, if there was a significance interaction of the factors analyzed, Tukey's post hoc analysis for the correction of multiple comparisons was applied. The central limit theorem states that with large sample sizes (n= >30), the sampling distribution of the mean tends to be normal, although if the data violate other assumptions this would not justify 100% use of parametric tests. (doi: 10.1146/annurev.publhealth.23.100901.140546.)

For variables that did not have a normal distribution (i.e., N<30), we used the Friedman test, which does not allow us to perform any correction analysis for multiple comparisons. The Friedman test is the most reliable alternative in terms of robustness to repeated-measures ANOVA when the assumptions of normality are not met. (doi.org/10.1201/9780429186196.)

The discussion makes some bold and causal physiological claims (e.g., HRC “promotes hypertrophy” or “reduces injury risk”) based solely on acute hormonal changes. There is clearly a choice of studies to support authors’ claims and no discussion of opposing findings (e.g., the debate on post-exercise humoral factors relevance – see PubMedID: 39190607).

Response: We thank the reviewer for the recommended article. We have added a new paragraph to the discussion section to address ambiguity in the subject, since muscle hypertrophy is driven by a complex system that is affected by internal and external factors.

The manuscript, especially discussion, is overwritten and repetitive. Entire paragraphs are copy-pasted textbook physiology with no connection to results.

Response: Thank you for your comment, but some biochemical and metabolic mechanisms cannot be summarized and sometimes have to be described literally.

Another major issue is the use of TBARS, which is nowadays considered almost inappropriate as a biomarker to evaluate oxidative stress. Sometimes called as T(hat) B(loody) A(ssay) (see PubMedID: 28371751).

Response: We understand your concern, for future studies we will use other biomarkers of oxidative stress.

Figure 3. Higher resolution in needed.

Response: Amended.

The concept of ‘a trend’ towards significance does not exist in statistics, although it is frequently mentioned by authors. No one knows if and how the results will change if we add observations.

Response: Thank you for your comment.  We use the concept “trend” based on Gravetter and Wallnau, who recognized the use of the concept trend in the description of results that are close to the threshold of statistical significance (p≥0.05), but do not rigorously reach it. These authors mention that a result that is not statistically significant does not necessarily mean that there is no effect; it may indicate a trend that did not reach significance due to insufficient sample size or data variability. They also warn that a trend should not be confused with a definitive conclusion and that using the concept “trend” without a statistical basis can lead to misinterpretation or over-interpretation of the data. Gravetter, F. J., & Wallnau, L. B. (2017). Statistics for the Behavioral Sciences (10th ed.). Cengage Learning.

The discussion on redox biology could be improved. I mean, with regards to exercise, mitochondria are not the main source of ROS. Instead, NADPH oxidases are the key molecule here.

Response: Thank you for your valuable comment. We agree that NADPH oxidases, particularly NOX2 and NOX4, have been identified as key contributors to reactive oxygen species (ROS) production in skeletal muscle during exercise, especially in response to metabolic stress.

However, we would like to note that the contribution of mitochondrial ROS cannot be entirely excluded, particularly during intense or prolonged exercise when mitochondrial oxygen consumption is substantially elevated. Recent literature suggests that the relative contribution of ROS sources may vary depending on exercise modality, intensity, and duration. For instance, mitochondrial ROS production has been shown to increase during sustained, high-intensity endurance exercise due to elevated electron flux through the electron transport chain, particularly at complex I and III.

To address this, we have revised the discussion, highlighting NADPH oxidases as major ROS producers during exercise while also acknowledging the context-dependent contribution of mitochondrial sources.

Author comment: We appreciate all the comments made on our manuscript, which helped improve it’s quality.

REVIEWER 1

The submitted manuscript aimed to examine the acute and recovery-phase biomarker responses (hormonal, inflammatory, oxidative stress, and muscle damage) to three types of resistance-based training: HIIT, CT, and HRC. Based on their findings, the authors report that HRC generates higher levels of testosterone than HIIT in the acute (P1) and late (P48) phases of recovery and produces lower levels of myostatin than CT and HIIT (P1) in the acute phase of recovery. The topic is valuable and always of interest, especially in an applied exercise and sports context. Please find next my comments that may be of help.

We thank the reviewer for his/her constructive and helpful feedback on our manuscript. We have replied to each comment in the section below and have introduced the corresponding edits into the manuscript using Word’s track changes.

The title and abstract refer to the “time course” of biomarker responses, yet much of the discussion fixates on comparing types of training. These are related but conceptually distinct Are the authors testing whether different protocols affect biomarkers differently or whether they follow similar temporal dynamics? Right now, it’s both — and neither is handled appropriately I think.

Response: We have modified our title to “Comparison of hormonal, inflammatory, muscle damage and oxidative stress biomarker changes in response to high-intensity interval, circuit and concurrent exercise bouts”.

The authors state that this is a crossover study in their methodology. Is it? Because in the results section they state that HRC (n = 12), HIIT (n = 10) and CT (n = 11). If it is a parallel-group study, is the sample size sufficient to ensure statistical power?

Response: The study was a crossover design and not a parallel-group study. The reason why sample sizes are different is because there were participants who were unable to attend the training day.

“Recreationally active” is a catch-all Please provide some information.

Response: Amended.

The results section is bloated with an excessive number of biomarkers and post hoc correlations. Many seem exploratory without correction for multiple comparisons (major issue – see next). Tables are hard to digest and lack clarity. For instance, repetitive listing of same baseline values in different groups is hard to follow.

Response: Thank you for your comment. In the statistical analysis of the methodology section, we explained that we used Tukey's post hoc analysis for the correction of multiple comparisons in ANOVA. Regarding the tables, we have grouped the parameters analyzed in several tables.

The authors bounce between ANOVA, Friedman, and dozens of Pearson/Spearman correlations seemingly at random. No correction for Type I error across ~30+ statistical tests. Please, apply corrections for multiple comparisons (e.g., Bonferroni or FDR) and you could also use heatmaps or summary graphs instead of overwhelming tables. If it is a crossover study, the whole statistical analysis approach should be revisited.

Response: Thank you for your comment. The selection criteria for the statistical method were based on the normality of the sample distribution and the number of data collected for each parameter (N). Therefore, we used ANOVA to analyze variables that did not have a normal distribution but exceeded 30 subjects measured for all measurement points. Furthermore, if there was a significance interaction of the factors analyzed, Tukey's post hoc analysis for the correction of multiple comparisons was applied. The central limit theorem states that with large sample sizes (n= >30), the sampling distribution of the mean tends to be normal, although if the data violate other assumptions this would not justify 100% use of parametric tests. (doi: 10.1146/annurev.publhealth.23.100901.140546.)

For variables that did not have a normal distribution (i.e., N<30), we used the Friedman test, which does not allow us to perform any correction analysis for multiple comparisons. The Friedman test is the most reliable alternative in terms of robustness to repeated-measures ANOVA when the assumptions of normality are not met. (doi.org/10.1201/9780429186196.)

The discussion makes some bold and causal physiological claims (e.g., HRC “promotes hypertrophy” or “reduces injury risk”) based solely on acute hormonal changes. There is clearly a choice of studies to support authors’ claims and no discussion of opposing findings (e.g., the debate on post-exercise humoral factors relevance – see PubMedID: 39190607).

Response: We thank the reviewer for the recommended article. We have added a new paragraph to the discussion section to address ambiguity in the subject, since muscle hypertrophy is driven by a complex system that is affected by internal and external factors.

The manuscript, especially discussion, is overwritten and repetitive. Entire paragraphs are copy-pasted textbook physiology with no connection to results.

Response: Thank you for your comment, but some biochemical and metabolic mechanisms cannot be summarized and sometimes have to be described literally.

Another major issue is the use of TBARS, which is nowadays considered almost inappropriate as a biomarker to evaluate oxidative stress. Sometimes called as T(hat) B(loody) A(ssay) (see PubMedID: 28371751).

Response: We understand your concern, for future studies we will use other biomarkers of oxidative stress.

Figure 3. Higher resolution in needed.

Response: Amended.

The concept of ‘a trend’ towards significance does not exist in statistics, although it is frequently mentioned by authors. No one knows if and how the results will change if we add observations.

Response: Thank you for your comment.  We use the concept “trend” based on Gravetter and Wallnau, who recognized the use of the concept trend in the description of results that are close to the threshold of statistical significance (p≥0.05), but do not rigorously reach it. These authors mention that a result that is not statistically significant does not necessarily mean that there is no effect; it may indicate a trend that did not reach significance due to insufficient sample size or data variability. They also warn that a trend should not be confused with a definitive conclusion and that using the concept “trend” without a statistical basis can lead to misinterpretation or over-interpretation of the data. Gravetter, F. J., & Wallnau, L. B. (2017). Statistics for the Behavioral Sciences (10th ed.). Cengage Learning.

The discussion on redox biology could be improved. I mean, with regards to exercise, mitochondria are not the main source of ROS. Instead, NADPH oxidases are the key molecule here.

Response: Thank you for your valuable comment. We agree that NADPH oxidases, particularly NOX2 and NOX4, have been identified as key contributors to reactive oxygen species (ROS) production in skeletal muscle during exercise, especially in response to metabolic stress.

However, we would like to note that the contribution of mitochondrial ROS cannot be entirely excluded, particularly during intense or prolonged exercise when mitochondrial oxygen consumption is substantially elevated. Recent literature suggests that the relative contribution of ROS sources may vary depending on exercise modality, intensity, and duration. For instance, mitochondrial ROS production has been shown to increase during sustained, high-intensity endurance exercise due to elevated electron flux through the electron transport chain, particularly at complex I and III.

To address this, we have revised the discussion, highlighting NADPH oxidases as major ROS producers during exercise while also acknowledging the context-dependent contribution of mitochondrial sources.

Author comment: We appreciate all the comments made on our manuscript, which helped improve it’s quality.

REVIEWER 1

The submitted manuscript aimed to examine the acute and recovery-phase biomarker responses (hormonal, inflammatory, oxidative stress, and muscle damage) to three types of resistance-based training: HIIT, CT, and HRC. Based on their findings, the authors report that HRC generates higher levels of testosterone than HIIT in the acute (P1) and late (P48) phases of recovery and produces lower levels of myostatin than CT and HIIT (P1) in the acute phase of recovery. The topic is valuable and always of interest, especially in an applied exercise and sports context. Please find next my comments that may be of help.

We thank the reviewer for his/her constructive and helpful feedback on our manuscript. We have replied to each comment in the section below and have introduced the corresponding edits into the manuscript using Word’s track changes.

The title and abstract refer to the “time course” of biomarker responses, yet much of the discussion fixates on comparing types of training. These are related but conceptually distinct Are the authors testing whether different protocols affect biomarkers differently or whether they follow similar temporal dynamics? Right now, it’s both — and neither is handled appropriately I think.

Response: We have modified our title to “Comparison of hormonal, inflammatory, muscle damage and oxidative stress biomarker changes in response to high-intensity interval, circuit and concurrent exercise bouts”.

The authors state that this is a crossover study in their methodology. Is it? Because in the results section they state that HRC (n = 12), HIIT (n = 10) and CT (n = 11). If it is a parallel-group study, is the sample size sufficient to ensure statistical power?

Response: The study was a crossover design and not a parallel-group study. The reason why sample sizes are different is because there were participants who were unable to attend the training day.

“Recreationally active” is a catch-all Please provide some information.

Response: Amended.

The results section is bloated with an excessive number of biomarkers and post hoc correlations. Many seem exploratory without correction for multiple comparisons (major issue – see next). Tables are hard to digest and lack clarity. For instance, repetitive listing of same baseline values in different groups is hard to follow.

Response: Thank you for your comment. In the statistical analysis of the methodology section, we explained that we used Tukey's post hoc analysis for the correction of multiple comparisons in ANOVA. Regarding the tables, we have grouped the parameters analyzed in several tables.

The authors bounce between ANOVA, Friedman, and dozens of Pearson/Spearman correlations seemingly at random. No correction for Type I error across ~30+ statistical tests. Please, apply corrections for multiple comparisons (e.g., Bonferroni or FDR) and you could also use heatmaps or summary graphs instead of overwhelming tables. If it is a crossover study, the whole statistical analysis approach should be revisited.

Response: Thank you for your comment. The selection criteria for the statistical method were based on the normality of the sample distribution and the number of data collected for each parameter (N). Therefore, we used ANOVA to analyze variables that did not have a normal distribution but exceeded 30 subjects measured for all measurement points. Furthermore, if there was a significance interaction of the factors analyzed, Tukey's post hoc analysis for the correction of multiple comparisons was applied. The central limit theorem states that with large sample sizes (n= >30), the sampling distribution of the mean tends to be normal, although if the data violate other assumptions this would not justify 100% use of parametric tests. (doi: 10.1146/annurev.publhealth.23.100901.140546.)

For variables that did not have a normal distribution (i.e., N<30), we used the Friedman test, which does not allow us to perform any correction analysis for multiple comparisons. The Friedman test is the most reliable alternative in terms of robustness to repeated-measures ANOVA when the assumptions of normality are not met. (doi.org/10.1201/9780429186196.)

The discussion makes some bold and causal physiological claims (e.g., HRC “promotes hypertrophy” or “reduces injury risk”) based solely on acute hormonal changes. There is clearly a choice of studies to support authors’ claims and no discussion of opposing findings (e.g., the debate on post-exercise humoral factors relevance – see PubMedID: 39190607).

Response: We thank the reviewer for the recommended article. We have added a new paragraph to the discussion section to address ambiguity in the subject, since muscle hypertrophy is driven by a complex system that is affected by internal and external factors.

The manuscript, especially discussion, is overwritten and repetitive. Entire paragraphs are copy-pasted textbook physiology with no connection to results.

Response: Thank you for your comment, but some biochemical and metabolic mechanisms cannot be summarized and sometimes have to be described literally.

Another major issue is the use of TBARS, which is nowadays considered almost inappropriate as a biomarker to evaluate oxidative stress. Sometimes called as T(hat) B(loody) A(ssay) (see PubMedID: 28371751).

Response: We understand your concern, for future studies we will use other biomarkers of oxidative stress.

Figure 3. Higher resolution in needed.

Response: Amended.

The concept of ‘a trend’ towards significance does not exist in statistics, although it is frequently mentioned by authors. No one knows if and how the results will change if we add observations.

Response: Thank you for your comment.  We use the concept “trend” based on Gravetter and Wallnau, who recognized the use of the concept trend in the description of results that are close to the threshold of statistical significance (p≥0.05), but do not rigorously reach it. These authors mention that a result that is not statistically significant does not necessarily mean that there is no effect; it may indicate a trend that did not reach significance due to insufficient sample size or data variability. They also warn that a trend should not be confused with a definitive conclusion and that using the concept “trend” without a statistical basis can lead to misinterpretation or over-interpretation of the data. Gravetter, F. J., & Wallnau, L. B. (2017). Statistics for the Behavioral Sciences (10th ed.). Cengage Learning.

The discussion on redox biology could be improved. I mean, with regards to exercise, mitochondria are not the main source of ROS. Instead, NADPH oxidases are the key molecule here.

Response: Thank you for your valuable comment. We agree that NADPH oxidases, particularly NOX2 and NOX4, have been identified as key contributors to reactive oxygen species (ROS) production in skeletal muscle during exercise, especially in response to metabolic stress.

However, we would like to note that the contribution of mitochondrial ROS cannot be entirely excluded, particularly during intense or prolonged exercise when mitochondrial oxygen consumption is substantially elevated. Recent literature suggests that the relative contribution of ROS sources may vary depending on exercise modality, intensity, and duration. For instance, mitochondrial ROS production has been shown to increase during sustained, high-intensity endurance exercise due to elevated electron flux through the electron transport chain, particularly at complex I and III.

To address this, we have revised the discussion, highlighting NADPH oxidases as major ROS producers during exercise while also acknowledging the context-dependent contribution of mitochondrial sources.

Author comment: We appreciate all the comments made on our manuscript, which helped improve it’s quality.

REVIEWER 1

The submitted manuscript aimed to examine the acute and recovery-phase biomarker responses (hormonal, inflammatory, oxidative stress, and muscle damage) to three types of resistance-based training: HIIT, CT, and HRC. Based on their findings, the authors report that HRC generates higher levels of testosterone than HIIT in the acute (P1) and late (P48) phases of recovery and produces lower levels of myostatin than CT and HIIT (P1) in the acute phase of recovery. The topic is valuable and always of interest, especially in an applied exercise and sports context. Please find next my comments that may be of help.

We thank the reviewer for his/her constructive and helpful feedback on our manuscript. We have replied to each comment in the section below and have introduced the corresponding edits into the manuscript using Word’s track changes.

The title and abstract refer to the “time course” of biomarker responses, yet much of the discussion fixates on comparing types of training. These are related but conceptually distinct Are the authors testing whether different protocols affect biomarkers differently or whether they follow similar temporal dynamics? Right now, it’s both — and neither is handled appropriately I think.

Response: We have modified our title to “Comparison of hormonal, inflammatory, muscle damage and oxidative stress biomarker changes in response to high-intensity interval, circuit and concurrent exercise bouts”.

The authors state that this is a crossover study in their methodology. Is it? Because in the results section they state that HRC (n = 12), HIIT (n = 10) and CT (n = 11). If it is a parallel-group study, is the sample size sufficient to ensure statistical power?

Response: The study was a crossover design and not a parallel-group study. The reason why sample sizes are different is because there were participants who were unable to attend the training day.

“Recreationally active” is a catch-all Please provide some information.

Response: Amended.

The results section is bloated with an excessive number of biomarkers and post hoc correlations. Many seem exploratory without correction for multiple comparisons (major issue – see next). Tables are hard to digest and lack clarity. For instance, repetitive listing of same baseline values in different groups is hard to follow.

Response: Thank you for your comment. In the statistical analysis of the methodology section, we explained that we used Tukey's post hoc analysis for the correction of multiple comparisons in ANOVA. Regarding the tables, we have grouped the parameters analyzed in several tables.

The authors bounce between ANOVA, Friedman, and dozens of Pearson/Spearman correlations seemingly at random. No correction for Type I error across ~30+ statistical tests. Please, apply corrections for multiple comparisons (e.g., Bonferroni or FDR) and you could also use heatmaps or summary graphs instead of overwhelming tables. If it is a crossover study, the whole statistical analysis approach should be revisited.

Response: Thank you for your comment. The selection criteria for the statistical method were based on the normality of the sample distribution and the number of data collected for each parameter (N). Therefore, we used ANOVA to analyze variables that did not have a normal distribution but exceeded 30 subjects measured for all measurement points. Furthermore, if there was a significance interaction of the factors analyzed, Tukey's post hoc analysis for the correction of multiple comparisons was applied. The central limit theorem states that with large sample sizes (n= >30), the sampling distribution of the mean tends to be normal, although if the data violate other assumptions this would not justify 100% use of parametric tests. (doi: 10.1146/annurev.publhealth.23.100901.140546.)

For variables that did not have a normal distribution (i.e., N<30), we used the Friedman test, which does not allow us to perform any correction analysis for multiple comparisons. The Friedman test is the most reliable alternative in terms of robustness to repeated-measures ANOVA when the assumptions of normality are not met. (doi.org/10.1201/9780429186196.)

The discussion makes some bold and causal physiological claims (e.g., HRC “promotes hypertrophy” or “reduces injury risk”) based solely on acute hormonal changes. There is clearly a choice of studies to support authors’ claims and no discussion of opposing findings (e.g., the debate on post-exercise humoral factors relevance – see PubMedID: 39190607).

Response: We thank the reviewer for the recommended article. We have added a new paragraph to the discussion section to address ambiguity in the subject, since muscle hypertrophy is driven by a complex system that is affected by internal and external factors.

The manuscript, especially discussion, is overwritten and repetitive. Entire paragraphs are copy-pasted textbook physiology with no connection to results.

Response: Thank you for your comment, but some biochemical and metabolic mechanisms cannot be summarized and sometimes have to be described literally.

Another major issue is the use of TBARS, which is nowadays considered almost inappropriate as a biomarker to evaluate oxidative stress. Sometimes called as T(hat) B(loody) A(ssay) (see PubMedID: 28371751).

Response: We understand your concern, for future studies we will use other biomarkers of oxidative stress.

Figure 3. Higher resolution in needed.

Response: Amended.

The concept of ‘a trend’ towards significance does not exist in statistics, although it is frequently mentioned by authors. No one knows if and how the results will change if we add observations.

Response: Thank you for your comment.  We use the concept “trend” based on Gravetter and Wallnau, who recognized the use of the concept trend in the description of results that are close to the threshold of statistical significance (p≥0.05), but do not rigorously reach it. These authors mention that a result that is not statistically significant does not necessarily mean that there is no effect; it may indicate a trend that did not reach significance due to insufficient sample size or data variability. They also warn that a trend should not be confused with a definitive conclusion and that using the concept “trend” without a statistical basis can lead to misinterpretation or over-interpretation of the data. Gravetter, F. J., & Wallnau, L. B. (2017). Statistics for the Behavioral Sciences (10th ed.). Cengage Learning.

The discussion on redox biology could be improved. I mean, with regards to exercise, mitochondria are not the main source of ROS. Instead, NADPH oxidases are the key molecule here.

Response: Thank you for your valuable comment. We agree that NADPH oxidases, particularly NOX2 and NOX4, have been identified as key contributors to reactive oxygen species (ROS) production in skeletal muscle during exercise, especially in response to metabolic stress.

However, we would like to note that the contribution of mitochondrial ROS cannot be entirely excluded, particularly during intense or prolonged exercise when mitochondrial oxygen consumption is substantially elevated. Recent literature suggests that the relative contribution of ROS sources may vary depending on exercise modality, intensity, and duration. For instance, mitochondrial ROS production has been shown to increase during sustained, high-intensity endurance exercise due to elevated electron flux through the electron transport chain, particularly at complex I and III.

To address this, we have revised the discussion, highlighting NADPH oxidases as major ROS producers during exercise while also acknowledging the context-dependent contribution of mitochondrial sources.

Author comment: We appreciate all the comments made on our manuscript, which helped improve it’s quality.

Reviewer 2 Report

Comments and Suggestions for Authors Dear Editor,

I am writing to submit my review of the manuscript titled “Time Course of Hormonal, Inflammatory, Muscle Damage and Oxidative Stress Biomarker Responses to High-Intensity Interval, Circuit and Concurrent Exercise Bouts.” The study addresses a relevant topic in sports physiology and contributes to the understanding of acute biochemical responses to different training modalities in recreationally active individuals. Please find the suggestions below.      

Page 2, Line 1–6 - Rephrase for clarity and precision.

Page 2, Line 7–13 - Consider citing authors to reflect ongoing research in testosterone-mediated muscle adaptation.

Page 2, Line 14–28 - Consider briefly summarizing how these hormonal responses translate into adaptations like strength or hypertrophy and simplifying technical phrasing.

Page 2, Line 45 ´- Add recent references (e.g., Radak et al., 2023) to strengthen the discussion on hormesis and oxidative stress adaptations.

Page 3, Line 26–43 - Include 1–2 references showing differences in endocrine or damage markers between modalities.

Page 3, Line 80 - Lack of a clearly defined hypothesis.

Page 4, Line 100 -  Add where participants were recruited (e.g., university, gym), and who performed recruitment.

Page 4, Line 107 -While ethics approval is mentioned, the trial registration is absent.
Page 5, Line 138- Clarify whether pre-exercise dietary control, hydration, or recent physical activity were standardized beyond the 3-day rest.

Page 6, Line 165 - Cite previous validation of treadmill protocol for VOâ‚‚max estimation.

Page 9, Line 290 - Break Table 1 into multiple tables (e.g., hormonal, inflammatory, oxidative markers) and standardize decimal places.

Page 11, Line 330 - Provide clearer figure legends and indicate statistical significance within the narrative (e.g., “as shown in Figure 1A”).

Page 12, Line 345 - Explain biological significance or theoretical justification for performing specific correlations.

Page 13, Line 400 - Compare findings with other 2020–2024 papers on post-exercise testosterone, myostatin, and oxidative markers (e.g., Ma et al., 2022, in J Appl Physiol).

Page 14, Line 435 - Update at least two references (e.g., replace Kraemer 2005 with a 2020+ review on anabolic response to circuit training).

Page 15, Line 488 - Add a sentence clarifying that this is an acute response and that chronic myostatin downregulation may differ.

Page 14, Line 14–30 – Testosterone Section - Provides a supported mechanistic interpretation of testosterone behavior.

Page 15, Line 1–25 – Clearly distinguish findings from literature vs your study. 

Page 15–16 – No clear distinction between mRNA vs protein in the cited studies. Your study measured plasma protein, while many studies referenced mRNA.

Page 16–17 – IL-6, TNF-α and TBARS - The paragraph becomes overly complex and speculative, particularly from Line 30 onward. Split into two sub-sections: 4.4 Inflammatory Response and 4.5 Oxidative Stress. Remove repetitive citations (e.g., Deminice, McBride), and focus on 3–4 key studies that best support your findings.

Author Response

REVIEWER 2

I am writing to submit my review of the manuscript titled “Time Course of Hormonal, Inflammatory, Muscle Damage and Oxidative Stress Biomarker Responses to High-Intensity Interval, Circuit and Concurrent Exercise Bouts.” The study addresses a relevant topic in sports physiology and contributes to the understanding of acute biochemical responses to different training modalities in recreationally active individuals. Please find the suggestions below.

 We thank the reviewer for their constructive and helpful feedback on our manuscript. We have replied to each comment in the section below and have introduced the corresponding edits into the manuscript using Word’s track changes.    

Page 2, Line 1–6 - Rephrase for clarity and precision.

Response: Amended.

Page 2, Line 7–13 - Consider citing authors to reflect ongoing research in testosterone-mediated muscle adaptation.

Response: Amended.

Page 2, Line 14–28 - Consider briefly summarizing how these hormonal responses translate into adaptations like strength or hypertrophy and simplifying technical phrasing.

Response: Linda

Page 2, Line 45 ´- Add recent references (e.g., Radak et al., 2023) to strengthen the discussion on hormesis and oxidative stress adaptations.

Response: Amended.

Page 3, Line 26–43 - Include 1–2 references showing differences in endocrine or damage markers between modalities.

Response: Amended.

Page 3, Line 80 - Lack of a clearly defined hypothesis.

Response: Amended.

Page 4, Line 100 -  Add where participants were recruited (e.g., university, gym), and who performed recruitment.

Response: Amended.
Page 4, Line 107 -While ethics approval is mentioned, the trial registration is absent.

Response: This research was not submitted to any clinical trial registry platform.

Page 5, Line 138- Clarify whether pre-exercise dietary control, hydration, or recent physical activity were standardized beyond the 3-day rest.

Response: Amended.

Page 6, Line 165 - Cite previous validation of treadmill protocol for VOâ‚‚max estimation.

Response: Amended

Page 9, Line 290 - Break Table 1 into multiple tables (e.g., hormonal, inflammatory, oxidative markers) and standardize decimal places.

Response: Following your suggestion, we have divided Table 1 into three tables. In relation to the decimal places, it is complicated to standardize them because of the difference in sensitivity of the parameter values.

Page 11, Line 330 - Provide clearer figure legends and indicate statistical significance within the narrative (e.g., “as shown in Figure 1A”).

Response: Amended.

Page 12, Line 345 - Explain biological significance or theoretical justification for performing specific correlations.

Response: Thank you for your comment. The purpose of correlations between the different parameters was to look for relationships between the markers analyzed at the different points measured.

Page 13, Line 400 - Compare findings with other 2020–2024 papers on post-exercise testosterone, myostatin, and oxidative markers (e.g., Ma et al., 2022, in J Appl Physiol).

Response: We have followed your suggestion, after performing a bibliographic search with the concepts you put in the statement, we have not found anything.

Page 14, Line 435 - Update at least two references (e.g., replace Kraemer 2005 with a 2020+ review on anabolic response to circuit training).

Response: Following your suggestion, we have checked line 435, but no reference to Kraemer 2005 appears.

Page 15, Line 488 - Add a sentence clarifying that this is an acute response and that chronic myostatin downregulation may differ.

Response: Amended.

Page 14, Line 14–30 – Testosterone Section - Provides a supported mechanistic interpretation of testosterone behavior.

Response: Amended.

Page 15, Line 1–25 – Clearly distinguish findings from literature vs your study. 

Response: Amended.

Page 15–16 – No clear distinction between mRNA vs protein in the cited studies. Your study measured plasma protein, while many studies referenced mRNA.

Response: We do not understand this issue well.

Page 16–17 – IL-6, TNF-α and TBARS - The paragraph becomes overly complex and speculative, particularly from Line 30 onward. Split into two sub-sections: 4.4 Inflammatory Response and 4.5 Oxidative Stress. Remove repetitive citations (e.g., Deminice, McBride), and focus on 3–4 key studies that best support your findings.

Response: Amended.

Author comment: We appreciate all the comments made on our manuscript, which helped improve it’s quality.

Round 2

Reviewer 1 Report

Comments and Suggestions for Authors

The authors are commended for their effort during the revision process.

No further comments from my side. 

Congrats 

Best regards 

Author Response

Thank you for your comment.